# Glucose to Platelet Ratio: A Potential Predictor of Hemorrhagic Transformation in Patients with Acute Ischemic Stroke

**DOI:** 10.3390/brainsci12091170

**Published:** 2022-08-31

**Authors:** Lingli Chen, Nan Chen, Yisi Lin, Huanzeng Ren, Qiqi Huang, Xiuzhen Jiang, Xiahui Zhou, Rongrong Pan, Wenwei Ren

**Affiliations:** 1Department of Neurology, The First Affiliated Hospital of Wenzhou Medical University, Wenzhou 325000, China; 2School of Nursing, Wenzhou Medical University, Wenzhou 325000, China; 3Department of Cardiac Care Unit, The First Affiliated Hospital of Wenzhou Medical University, Wenzhou 325000, China; 4Department of Rehabilitation Medicine, Pingyang Changgung Yining Hospital, Wenzhou 325000, China; 5Drug Clinical Trial Organization, Wenzhou Traditional Chinese Medicine Hospital, Wenzhou 325000, China; 6Department of Rehabilitation Medicine, Wenzhou Traditional Chinese Medicine Hospital, Wenzhou 325000, China

**Keywords:** stroke, hemorrhagic transformation, glucose, platelet, ratio

## Abstract

Glucose and platelet are two easily obtained clinical indicators; the present research aimed to demonstrate their association with hemorrhagic transformation (HT) in acute ischemic stroke (AIS) patients without thrombolytic or thrombectomy therapy. This was a single-center retrospective study. Patients who were diagnosed with HT after AIS were included in the HT group. Meanwhile, using the propensity score matching (PSM) approach, with a ratio of 1:2, matched patients without HT were included in the non-HT group. Serum G/P levels were measured on the first morning after admission (at least eight hours after the last meal). Characteristics were compared between the two groups. Multivariate logistic regression was used to determine the independent relationship between G/P and HT after AIS, with G/P being divided into quartiles. From January 2013 to March 2022, we consecutively included 643 AIS patients with HT (426/643 [66.25%] with HI and 217/643 [33.75%] with PH), and 1282 AIS patients without HT, at the First Affiliated Hospital of Wenzhou Medical University. The HT group had higher G/P levels than the non-HT group (0.04 ± 0.02 vs. 0.03 ± 0.02, *p* < 0.001). However, there was no difference in G/P levels between HI and PH subgroups (0.04 ± 0.02 vs. 0.04 ± 0.02, *p* > 0.05). Moreover, the G/P levels were divided into quartiles (Q1 ≤ 0.022; Q2 = 0.023–0.028; Q3 = 0.029–0.039; Q4 ≥ 0.040), with Q1 being settled as the reference layer. After controlling the confounders, multivariate regression analyses showed that the Q4 layer (Q4: G/P ≥ 0.040) was independently associated with elevated HT risk (odds ratio [OR] = 1.85, 95% CI = 1.31–2.63, *p* < 0.001). G/P levels on admission were independently associated with HT risk in AIS patients. In clinical practice, adequate attention should be paid to AIS patients with elevated G/P levels (G/P ≥ 0.040).

## 1. Introduction

Acute ischemic stroke (AIS), a major cerebrovascular disorder with high morbidity, disability, and mortality, is caused by a cerebral artery blockage that inevitably results in a lack of oxygenation, ultimately leading to localized neurological impairments and irreversible clinical symptoms, threatening human health and life [1,2]. In China, AIS’s societal and familial cost has grown significantly in recent decades [3]. Hemorrhagic transformation (HT), which can be classified as hemorrhagic infarction (HI) and parenchymatous hematoma (PH), based on the European cooperative acute stroke study (ECASS II), is one of the most common and severe complications of AIS. HT may be triggered by either the normal progression of AIS or reperfusion treatment, often resulting in poor clinical outcomes [4,5]. Therefore, early identification of patients at high risk of HT is crucial.

Risk factors related to HT include massive cerebral infarction, cerebral cortex infarction, atrial fibrillation (AF), low cholesterol levels, higher National Institute of Health Stroke Scale (NIHSS) scores, poor collateral vessels, application of intravenous recombinant tPA, and early CT signs [4,6]. Previous studies have shown that glucose levels and platelet counts were also associated with HT [7,8,9,10,11]. Blood glucose levels can reflect physiological stress, metabolism, and nutritional status [12,13,14]. In experimental middle cerebral artery occlusion models with reperfusion, high glucose levels induced HT [15,16]. A prospective cohort study showed increased glucose levels predisposed to HT [17]. Acutely elevated glucose levels, regardless of a previous diagnosis of diabetes, were a significant predictor of HT in patients with AIS [18]. Other studies observed that low glucose levels were also linked to HT [8,19]. In the first 36 h following successful endovascular recanalization treatment, time-related glycemic variability was strongly correlated with symptomatic intracerebral hemorrhage and poor functional outcomes [8]. Due to their hemostatic function and their significant participation in the inflammatory response, platelets have been regarded as a crucial factor against HT during ischemia/reperfusion [4,11]. A prospective multicenter intravenous thrombolysis register-based study suggested that a lower platelet count was related to an increased risk of symptomatic intracranial hemorrhage. By comparison, a higher platelet count indicated increased mortality [20]. Given that the roles of glucose and platelet count in HT remain controversial, we need to find a new indicator that combines serum glucose and platelet count to replace them as a more accurate predictor for HT.

The association between the glucose-to-platelet ratio (G/P) and HT in AIS patients has not been thoroughly investigated. Previous studies have investigated the risk factors of HT in patients with thrombolytic or thrombectomy therapy, but few have investigated the risk factors of HT in patients without thrombolytic or thrombectomy treatment [21]. This study aimed to determine if G/P was connected with HT independently in patients without thrombolytic or thrombectomy therapy.

## 2. Materials and Methods

### 2.1. Study Population

As shown in Figure 1, this study included consecutive AIS patients from the department of neurology in the First Affiliated Hospital of Wenzhou Medical University, from January 2013 to March 2022. Patients were included in this retrospective study if they met all the following inclusion criteria: (1) ≥18 years old; (2) hospitalized within one week from the onset of AIS; (3) confirmed diagnosis of AIS with brain computerized tomography (CT) or brain magnetic resonance imaging (MRI). Patients were excluded if they met one of the following exclusion criteria: (1) were diagnosed with transient ischemic attack (TIA); (2) received thrombolytic or thrombectomy therapy; (3) were diagnosed with cerebral venous sinus thrombosis (CVST); (4) patients’ medical records were incomplete.

The research was approved by the ethics committee of the First Affiliated Hospital of Wenzhou Medical University with the registration number KY2021-R077, and complied with the local Research Ethics Committee’s ethical criteria on human experimentation. Every procedure followed the Helsinki Declaration. Although we could not gain the patients’ informed consent, owing to the use of a retrospective research design, we were granted permission to gather data from our stroke registry.

### 2.2. Data Collection

We collected and documented the demographic data (age and gender), lifestyle risk factors for stroke (drinking and smoking history), baseline clinical parameters (body mass index [BMI], hypertension, diabetes mellitus, AF), laboratory information (glucose levels, platelet count, creatine, prothrombin time [PT], and international normalized ratio [INR]), treatment during hospitalization (antiplatelet, anticoagulation, and statins), the time interval between the onset of stroke and baseline scans, and the time interval between the onset of stroke and follow-up scans. Blood samples were measured on the first morning after admission (at least eight hours after the last meal). The severity of AIS was evaluated by the NIHSS score at admission. All patients were categorized into large artery atherosclerosis, cardioembolism, small vessel occlusion, and other subtypes (including stroke of other determined etiology and stroke of undetermined etiology), with the trial of ORG 10,172 in acute stroke treatment (TOAST) criteria [22].

Infarct locations were classified as the following subtypes, based on brain scans (MRI or CT): lobar (frontal, temporal, parietal, occipital, insular); subcortical (corona radiata, thalamus, internal capsule, basal ganglia, corpus callosum); brainstem (medulla, pons, midbrain); cerebellum; and mixed type (contained at least two of the subtypes listed above).

According to the electronic medical records, brain CT scans (including CT scans from our hospital and other hospitals’ neurology outpatient or emergency departments) were performed on all patients before admission. HT was identified, by utilizing repeat CT/MRI scans during hospitalization conventionally or at times when there was any sign of clinical deterioration. Patients without a second CT/MRI scan were excluded. Two neuroimaging doctors evaluated the MRI/CT scans. According to the ECASS II categorization guidelines, HT was divided into HI and PH [23]. CT scan examples of HI and PH are shown in Figure 2.

### 2.3. Statistical Analyses

The normally distributed continuous variables were shown as the mean ± standard deviation (SD), while the non-normally distributed continuous variables were exhibited as the median and interquartile range (IQR). Meanwhile, categorical variables were summarized using numbers and percentages. The Student’s test and Analysis of Variance were used to analyze the differences between normally distributed continuous variables. The Mann–Whitney and Kruskal–Wallis tests were used, to analyze the differences between no-normally distributed continuous variables. The categorical variables were compared, using Chi-square and Fisher’s exact tests. Multivariable logistic regression analysis was conducted to investigate the connection between HT and clinical features. In addition, we compared the differences in G/P levels among the HI, PH, and non-HT groups, with Analysis of Variance, and Bonferroni correction was used in the post-hoc analysis. Then, a bar chart was drawn to demonstrate the differences. A forest plot was created, after using multivariate logistic regression models to examine the effect of G/P on HT when controlling confounders (including BMI, hypertension history, AF history, NIHSS score, anticoagulant, antiplatelet, statins, TOAST, and infarct location). We utilized the propensity score matching (PSM) approach, with a ratio of 1:2, to include patients with HT, and age and gender-matched patients without HT. All statistical analyses were performed in R version 4.1.3 (R Foundation for Statistical Computing, Vienna, Austria, 2022). Our graphical abstract was drawn using Figdraw (Hangzhou Duotai Technology Co., Ltd., Hangzhou, China, 2022).

## 3. Results

### 3.1. Characteristics of Enrolled Patients

As shown in Figure 1, we reviewed 4757 consecutive patients; 1208 patients were excluded, based on the above exclusion criteria, and 3549 patients (643 individuals with HT and 2906 individuals without HT) were initially included. Using the PSM approach, we finally recruited 643 patients with HT, and 1282 age and gender-matched patients without HT, in a ratio of 1:2.

The baseline characteristics of patients with and without HT were presented in Table 1. The levels of G/P were significantly higher in HT patients compared to those without HT (0.04 ± 0.02 vs. 0.03 ± 0.02, *p* < 0.001). HT patients had higher BMI levels, history of hypertension, history of AF, higher NIHSS scores, higher glucose levels, and lower platelet counts (all *p* < 0.05). They were more likely to undergo anticoagulants, and less likely to undergo antiplatelet or statin (all *p* < 0.05). Patients with cardioembolic cerebral infarction had a higher HT risk than patients with non-cardioembolic cerebral infarction (*p* < 0.001). Regarding infarct location, brainstem or subcortical infarcts were less prone to undergo HT (all *p* < 0.05). The results showed that patients with HT had a longer time interval between the onset of stroke and follow-up scans than those without HT (5 [3,4,5,6,7] vs. 7 [5,6,7,8,9,10,11], *p* < 0.001). In addition, there were no significant differences between drinking history, smoking history, diabetes, creatinine, PT, and INR (all *p* > 0.05).

### 3.2. Characteristics of HI and PH Patients

Among the HT patients, 426 (426/643, 66.25%) were diagnosed with HI, and 217 (217/643, 33.75%) were diagnosed with PH. The baseline characteristics of non-HT, HI, and PH patients are presented in Table 2. There was no difference in G/P levels between the HI and PH subgroups (0.04 ± 0.02 vs. 0.04 ± 0.02, *p* > 0.05) (Figure 3). There were substantial differences in BMI, hypertension, NIHSS scores, glucose levels, platelet counts, anticoagulants, antiplatelet, statin, TOAST criteria, time interval ^a^, time interval ^b^, and infarct location in non-HT, HI, and PH groups (all *p* < 0.05). At the same time, there were no significant differences in drinking history, smoking history, diabetes, creatinine, PT, and INR in the three groups (all *p* > 0.05).

### 3.3. Multivariable Logistic Regression Analysis of the Association between Variables and the HT Risk

We conducted a multivariate logistic regression analysis on patients to evaluate the relationship between HT patients and risk factors. The forest plot showed the results (Figure 4). The G/P levels were divided into quartiles (Q1 ≤ 0.022, Q2 = 0.023–0.028, Q3 = 0.029–0.039, Q4 ≥ 0.040), with Q1 being settled as the reference layer. After controlling the confounders, multivariate regression analyses showed that the Q4 layer (Q4:G/P ≥ 0.040) was independently associated with elevated HT risk (OR = 1.85, 95%, CI = 1.31–2.63, *p* < 0.001). Meanwhile, higher NIHSS scores (OR: 1.05, 95%, CI = 1.02–1.09, *p* = 0.004), higher BMI (OR: 1.06, 95%, CI = 1.01–1.12, *p* = 0.012), longer time interval between the onset of stroke and follow-up scans (OR: 1.12, 95%, CI = 1.09–1.14, *p* < 0.001), history of AF (OR: 1.99, 95%, CI = 1.24–3.19, *p* = 0.005) and cerebellum stroke (OR: 2.37, 95%, CI = 1.35–4.15, *p* = 0.011) were also independent risk factors for HT. Patients who suffered HT were less likely to receive antiplatelet during hospitalization (OR: 0.62, 95%, CI = 0.43–0.90, *p* = 0.011).

After adjusting confounding variables, when G/P was introduced as a continuous variable to the multivariate logistic regression analysis, similar results were obtained (*p* < 0.05). Furthermore, we performed a receiver operating characteristic (ROC) curve, to compare the predictive ability of the G/P, glucose, and platelet. The results (Appendix A) showed that G/P had the best predictive value (AUC: 0.616, 95% CI 0.594–0.637), superior to both glucose (AUC:0.594, 95% CI 0.571–0.616) and platelet (AUC: 0.563, 95% CI 0.541–0.585) (both *p* < 0.05).

## 4. Discussion

HT often leads to poor outcomes in AIS [4,6]. To strike a balance between risks and benefits, it is crucial to identify influential prognostic factors for patients at high HT risk. To the best of our knowledge, this is the first research to investigate the connection between G/P and HT in patients with AIS. According to the present study, high G/P levels (G/P ≥ 0.040) were independently related to increased incidence of HT following AIS, and there was no difference in G/P levels between HI and PH subtypes.

A substantial amount of research has been devoted to reducing the risk of HT in AIS to enhance patients’ outcomes [6]. A high glucose level has been identified as a risk factor for HT, particularly for symptomatic intracranial hemorrhage and PH [17,24]. When blood glucose levels exceed 150 mg/dL, the risk of HT triples, compared to normoglycemia [17]. In addition, several studies have linked high glucose levels to increased death rates and poorer functional results [25]. Animal models have also shown this relationship [26,27]. Consistent with previous studies, this study showed that the glucose levels in HT patients were significantly higher than those in non-HT patients [17]. Several theories may explain the reported connection between high glucose levels and HT, even if the underlying processes are not well known. Firstly, high glucose levels increase endothelial dysfunction, leading to blood–brain barrier damage, increasing the HT risk in the case of AIS [17,28]. Secondly, massive oxidative stress is created by high glucose levels and ischemia/reperfusion damage, severely compromising the blood–brain barrier, and resulting in HT [28,29,30]. Thirdly, inflammation is exacerbated by the production of proinflammatory cytokines, apoptosis, and the exacerbation of cytotoxic edema when glucose levels are high [28,31]. Inflammation plays a vital role in HT. The neutrophil-to-lymphocyte ratio can accurately predict symptomatic hemorrhagic transformation in AIS patients who undergo revascularization [32]. AIS patients having endovascular treatment, and achieving effective recanalization with a higher systemic inflammatory response index at admission, had an increased risk of a poor functional outcome at three months [33]. Early ficolin-1 is one of the most sensitive predictors of functional prognosis in AIS [34]. Furthermore, there is increasing evidence that readily available serum biomarkers of inflammation can also be reliable predictors of outcomes in patients with ICH, and improve the outcome prediction when added to validated prognostic scales. C-reactive protein, an important inflammatory marker, is associated with intracerebral hemorrhage outcomes [35]. Matrix metalloproteinases have a pleiotropic and biphasic effect on acute intracerebral hemorrhage [36]. In addition, the neutrophil-to-lymphocyte ratio is associated with 30-day functional status after acute intracerebral hemorrhage [37]. On the other hand, the literature has also shown that low glucose levels may lead to HT in AIS patients through numerous possible routes, including releasing inflammatory markers, acute hypertensive response, and platelet activation [8,19].

Platelets—tiny blood cells renowned for their traditional function in hemostasis—have a crucial role in inflammation, angiogenesis, and regulated apoptosis after tissue injury [38,39]. The function of platelets as a predictor of HT following thrombolysis has been studied, but the findings were contradictory [40]. According to a prior study, low platelet counts did not substantially raise the risk of HT, and waiting for platelet counts caused an unsubstantiated delay in intravenous thrombolysis therapy [41]. By contrast, another study suggested that lower baseline platelet counts were linked to higher HT risk after intravenous thrombolysis. Clinical guidelines published in 2018 by the American Heart Association/American Stroke Association did not recommend reperfusion therapy in patients with platelets <100,000/mm^3^ [20,42]. Low baseline platelet counts were related to an increased risk of HT after AIS without thrombolysis, according to the present study.

This study suggested that higher G/P levels (G/P ≥ 0.04) were independently related to increased HT risk. In addition, G/P had the best predictive value when compared with glucose and platelets. Consistent with previous research, we also found that patients with higher BMI levels, higher NIHSS scores, history of AF, and location in the cerebellum were more likely to suffer HT [6,29].

This research has certain limitations that should be noted. Firstly, as this was a retrospective single-center study, cause-and-effect linkages could not be determined, and prospective multicenter trials are required, to prove the causation and give more trustworthy long-term prognostic information. Secondly, the G/P was only recorded once, and the relationship between dynamic changes in G/P and HT should be studied in the future. Thirdly, we did not explore the underlying mechanism between G/P and HT in animal models. Fourthly, patients with thrombolytic or thrombectomy treatment in this study were excluded. In the future, we will investigate the relationship between G/P and HT in patients with thrombolytic or thrombectomy therapy, with a larger sample. Fifthly, we analyzed the time interval between the onset of symptoms and scans (baseline and follow-up). The results showed that patients with HT had a longer time interval between baseline and follow-up imaging than those without HT, which may have led to bias. Finally, although we routinely repeated CT or MRI scans, we still could not completely confirm that all the patients with asymptomatic hemorrhagic were identified in this study, which may have led to bias. Thus, more rigorous prospective studies are needed in the future.

## 5. Conclusions

In conclusion, G/P may function as a biomarker for HT in AIS patients. In clinical practice, AIS patients with high G/P levels (G/P ≥ 0.040) should get more attention, to reduce the risk of HT.

## Figures and Tables

**Figure 1 brainsci-12-01170-f001:**
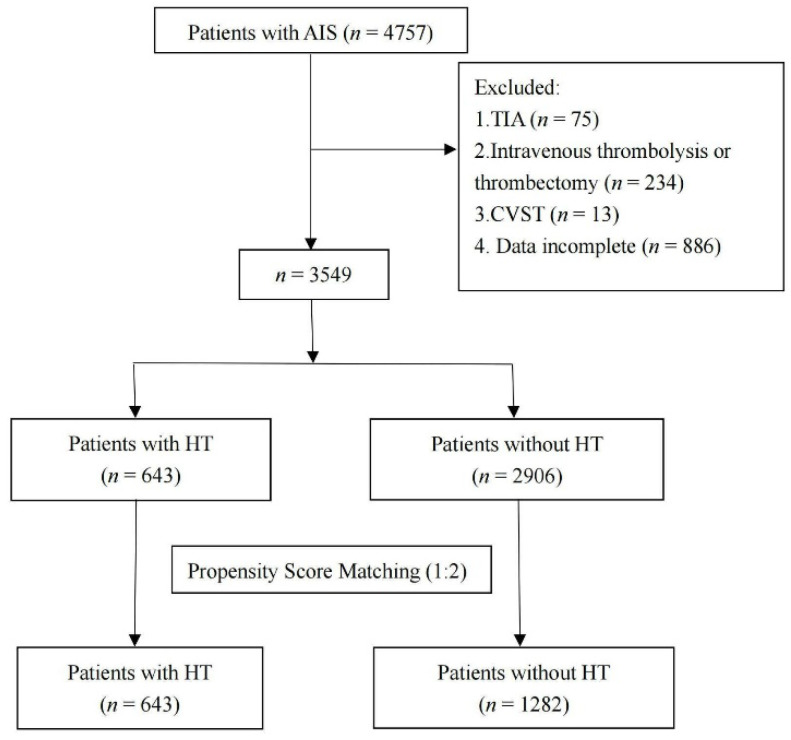
Flow chart showing the patient’s selection process. AIS, acute ischemic stroke; TIA, transient ischemic attack; CVST, cerebral venous sinus thrombosis; HT, hemorrhagic transformation.

**Figure 2 brainsci-12-01170-f002:**
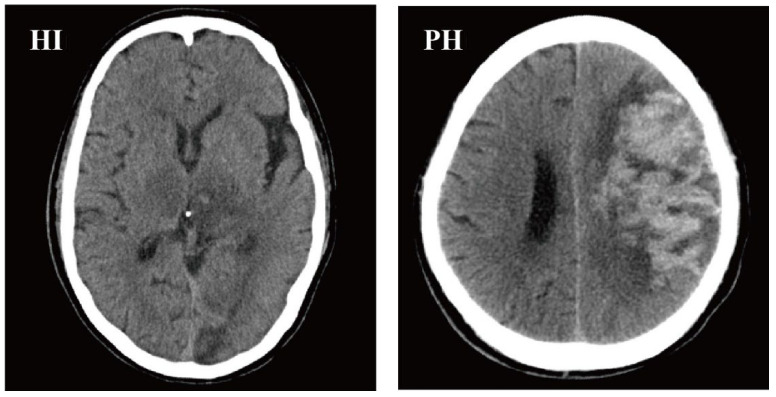
CT scan examples of HI and PH, according to the ECASS II classification. HI, hemorrhagic infarction; PH, parenchymal hemorrhage; ECASS II, European Cooperative Acute Stroke Study II.

**Figure 3 brainsci-12-01170-f003:**
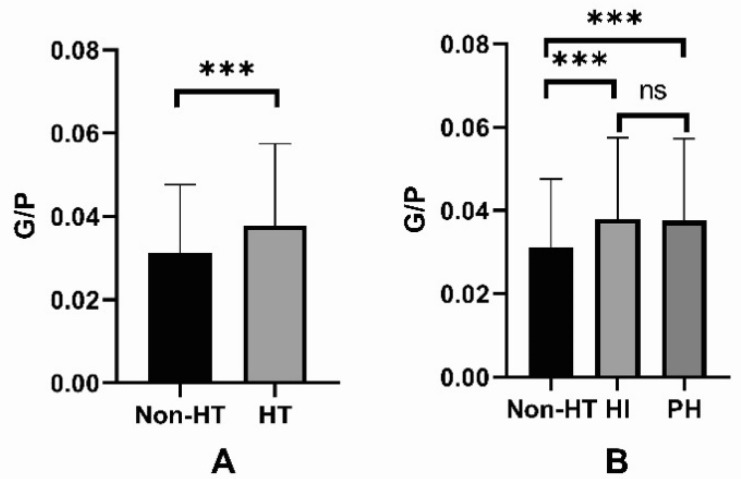
(**A**) The levels of G/P in the non-HT and HT groups; (**B**) The levels of G/P in the non-HT, HI, and PH groups. G/P, glucose to platelet ratio; HT, hemorrhagic transformation; HI, hemorrhagic infarction; PH, parenchymal hemorrhage. The ns stands for not statistically significant. *** *p* < 0.001.

**Figure 4 brainsci-12-01170-f004:**
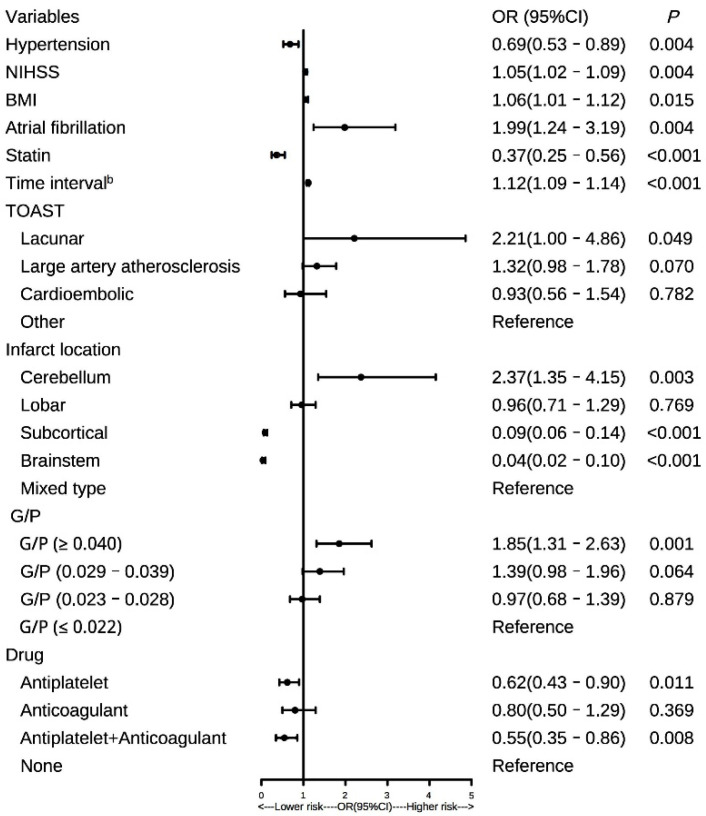
Forest plot of the relationship between variables and the HT risk in patients with AIS. G/P, glucose to platelet ratio; HT, hemorrhagic transformation; AIS, acute ischemic stroke; BMI, body mass index; AF, atrial fibrillation; NIHSS, National Institutes of Health Stroke Scale; Time interval ^b^, time from stroke onset to follow-up scans; TOAST, the trial of ORG 10,172 in acute stroke treatment; OR, odds ratio.

**Table 1 brainsci-12-01170-t001:** Characteristics of AIS patients with or without HT.

Variables	Non-HT (*n* = 1282)	HT (*n* = 643)	Statistic	*p*
Age (years), Mean ± SD	69.64 ± 11.85	69.57 ± 11.9	0.114	0.909
Gender, *n* (%)			0.06	0.806
Male	858 (66.93)	426 (66.25)		
Female	424 (33.07)	217 (33.75)		
BMI (kg/m^2^), Mean ± SD	23.72 ± 2.59	24.36 ± 7.57	−2.061	0.04
Drinking, *n* (%)	393 (30.66)	216 (33.59)	1.575	0.209
Smoking, *n* (%)	467 (36.43)	257 (39.97)	2.14	0.143
Hypertension, *n* (%)	919 (71.68)	416 (64.7)	9.512	0.002
Diabetes, *n* (%)	360 (28.08)	191 (29.7)	0.476	0.49
AF, *n* (%)	209 (16.3)	266 (41.37)	143.41	<0.001
NIHSS, Median (Q1, Q3)	2 (1, 6)	3 (1, 8)	338,743.5	<0.001
Time interval ^a^ (days), Median (Q1, Q3)	0 (0–1)	0 (0–1)	1.409	0.159
Time interval ^b^ (days), Median (Q1, Q3)	5 (3–7)	7 (5–11)	11.874	<0.001
TOAST classification, *n* (%)			129.742	<0.001
Large artery atherosclerosis	449 (35.02)	184 (28.62)		
Small vessel occlusion	47 (3.67)	16 (2.49)		
Cardioembolism	204 (15.91)	251 (39.04)		
Other	582 (45.4)	192 (29.86)		
Infarct location, *n* (%)			448.145	<0.001
Lobar	174 (13.57)	158 (24.57)		
Subcortical	570 (44.46)	43 (6.69)		
Brainstem	164 (12.79)	6 (0.93)		
Cerebellum	25 (1.95)	50 (7.78)		
Mixed type	349 (27.22)	386 (60.03)		
Drugs, *n* (%)			161.61	<0.001
None	101 (7.88)	146 (22.71)		
Antiplatelet	952 (74.26)	309 (48.06)		
Anticoagulant	75 (5.85)	98 (15.24)		
Antiplatelet + Anticoagulant	154 (12.01)	90 (14)		
Statin, *n* (%)	1202 (93.76)	522 (81.18)	71.106	<0.001
Creatinine (umol/L), Mean ± SD	81.29 ± 61.23	79.21 ± 55.09	0.751	0.453
PT, Median (Q1, Q3)	13.8 (13.2, 14.3)	13.9 (13.3, 14.3)	393,247	0.1
INR, Median (Q1, Q3)	1.07 (1.01, 1.12)	1.08 (1.02, 1.11)	397,520	0.202
Glucose (mmol/L), Mean ± SD	6.17 ± 2.61	7.03 ± 3.26	−5.806	<0.001
Platelet (10^9^/L), Mean ± SD	215.09 ± 68.66	200.52 ± 62.57	4.663	<0.001
G/P, Mean ± SD	0.03 ± 0.02	0.04 ± 0.02	−7.338	<0.001

Notes: AIS, acute ischemic stroke; HT, hemorrhagic transformation; BMI, body mass index; AF, atrial fibrillation; NIHSS, Institutes of Health Stroke Scale; Time interval ^a^, time from stroke onset to baseline scans; Time interval ^b^, time from stroke onset to follow-up scans; TOAST, the trial of ORG 10,172 in acute stroke treatment; PT, prothrombin time; INR, international normalized ratio; G/P, glucose to platelet ratio.

**Table 2 brainsci-12-01170-t002:** Characteristics of patients with HI or with PH.

Variables	Non-HT (*n* = 1282)	HI (*n* = 426)	PH (*n* = 217)	Statistic	*p*
Age (years), Mean ± SD	69.64 ± 11.85	69.13 ± 11.98	70.45 ± 11.72	0.893	0.409
Gender, *n* (%)				2.211	0.331
Male	858 (66.93)	274 (64.32)	152 (70.05)		
Female	424 (33.07)	152 (35.68)	65 (29.95)		
BMI (kg/m^2^), Mean ± SD	23.72 ± 2.59	24.63 ± 9.18	23.82 ± 2.03	5.608	0.004
Drinking, *n* (%)	393 (30.66)	136 (31.92)	80 (36.87)	3.331	0.189
Smoking, *n* (%)	467 (36.43)	165 (38.73)	92 (42.4)	3.111	0.211
Hypertension, *n* (%)	919 (71.68)	280 (65.73)	136 (62.67)	10.469	0.005
Diabetes, *n* (%)	360 (28.08)	144 (33.8)	47 (21.66)	10.93	0.004
AF, *n* (%)	209 (16.3)	167 (39.2)	99 (45.62)	147.944	<0.001
NIHSS, Median (Q1, Q3)	2 (1, 6)	3 (1, 8)	4 (2, 8)	47.214	<0.001
Time interval ^a^ (days), Median (Q1, Q3)	0 (0–1)	0 (0–2)	0 (0–1)	2.514	0.284
Time interval ^b^ (days), Median (Q1, Q3)	5 (3–7)	7 (5–11)	7 (5–12)	141.13	<0.001
TOAST classification, *n* (%)				143.875	<0.001
Large artery atherosclerosis	449 (35.02)	135 (31.69)	49 (22.58)		
Small vessel occlusion	47 (3.67)	10 (2.53)	6 (2.76)		
Cardioembolism	204 (15.91)	148 (34.74)	103 (47.47)		
Other	582 (45.4)	133 (31.22)	49 (22.58)		
Infarct location, *n* (%)				451.211	<0.001
Lobar	174 (13.57)	111 (26.06)	47 (21.66)		
Subcortical	570 (44.46)	31 (7.28)	12 (5.53)		
Brainstem	164 (12.79)	4 (0.94)	2 (0.92)		
Cerebellum	25 (1.95)	32 (7.51)	18 (8.29)		
Mixed type	349 (27.22)	248 (58.22)	138 (63.59)		
Drugs, *n* (%)				188.508	<0.001
None	101 (7.88)	76 (17.84)	70 (32.26)		
Antiplatelet	952 (74.26)	222 (52.11)	87 (40.09)		
Anticoagulant	75 (5.85)	67 (15.73)	31 (14.29)		
Antiplatelet + Anticoagulant	154 (12.01)	61 (14.32)	29 (13.36)		
Statin, *n* (%)	1202 (93.76)	357 (83.8)	165 (76.04)	81.717	<0.001
Creatinine (umol/L), Mean ± SD	81.29 ± 61.23	80.38 ± 65.64	76.91 ± 23.13	0.509	0.601
PT, Median (Q1, Q3)	13.8 (13.2, 14.3)	13.9 (13.3, 14.3)	13.97 (13.4, 14.3)	4.285	0.117
INR, Median (Q1, Q3)	1.07 (1.01, 1.12)	1.08 (1.02, 1.11)	1.08 (1.02, 1.11)	2.204	0.332
Glucose (mmol/L), Mean ± SD	6.17 ± 2.61	7.13 ± 3.37	6.84 ± 3.04	20.203	<0.001
Platelet (10^9^/L), Mean ± SD	215.09 ± 68.66	202.11 ± 63.8	197.4 ± 60.13	10.58	<0.001
G/P, Mean ± SD	0.03 ± 0.02	0.04 ± 0.02	0.04 ± 0.02	30.251	<0.001

Notes: AIS, acute ischemic stroke; HT, hemorrhagic transformation; BMI, body mass index; AF, atrial fibrillation; NIHSS, Institutes of Health Stroke Scale; Time interval ^a^, time from stroke onset to baseline scans; Time interval ^b^, time from stroke onset to follow-up scans; TOAST, the trial of ORG 10,172 in acute stroke treatment; PT, prothrombin time; INR, international normalized ratio; G/P, glucose to platelet ratio.

## Data Availability

The raw data supporting the conclusions of this article will be made available by the authors, without undue reservation.

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
