# Peer review of "Glucose to Platelet Ratio: A Potential Predictor of Hemorrhagic Transformation in Patients with Acute Ischemic Stroke"

_brainsci, 2022, doi:10.3390/brainsci12091170_

Round 1
Reviewer 1 Report
The paper is a retrospective study looking at association between glucose /platelet index and haemorrhagic changes in stroke. Bleeding is a serious, potentially fatal complication of ischemic stroke, and is extensively studied. Identification of risk factors is particularly important when considering revascularisation therapy. Both high glucose level and low platelets increase the risk of bleeding, the authors proposed a new predictor, the glucose platelet index to assess risk.
The research is well conducted and presented. However, there are issues with study design that undermines the value of the work
1. The control group is matched for age and sex. However, the two groups are not balanced for well known factors for haemorrhagic transformation, particularly hypertension, diabetes, and use of anti-platelet or anticoagulation.
2. The study excluded patients who received revascularisation therapy, This group is the focus of majority of previous studies because it is the group at highest risk, and in which a predictor of haemorrhage can potentially change treatment decision. The authors did not clarify why they excluded this group.
3. The authors did not clarify why they think the glucose/platelet index is better than glucose level or platelet count alone. In the logistic regression, these factors
Other minor comments
Line 101: smoking and alcohol are lifestyle risk factors for stroke, but not usually referred to as hobbies
Line 118: Baseline imaging is reported to be done at a different department. Why is that? And is there any further information on time between onset of symptoms and scans (baseline and follow up). I note that the study only included patients were a repeat scan was done for clinical reasons. This might be a source of bias as ICH is often asymptomatic. Is it clear whether the heamorrhagic changes were always symtomatic
Author Response
Responses to the Reviewer 1:
Question 1: The paper is a retrospective study looking at association between glucose /platelet index and haemorrhagic changes in stroke. Bleeding is a serious, potentially fatal complication of ischemic stroke, and is extensively studied. Identification of risk factors is particularly important when considering revascularisation therapy. Both high glucose level and low platelets increase the risk of bleeding, the authors proposed a new predictor, the glucose platelet index to assess risk. The research is well conducted and presented. However, there are issues with study design that undermines the value of the work. The control group is matched for age and sex. However, the two groups are not balanced for well known factors for haemorrhagic transformation, particularly hypertension, diabetes, and use of anti-platelet or anticoagulation.
Answer: Thanks for your valuable suggestions. We appreciate it very much! These factors were not matched since we aimed to identify how these factors affected hemorrhagic transformation. In addition, the logistic regression model could also control these variables to highlight the relationship between the G/P ratio and HT.
Question 2: The study excluded patients who received revascularisation therapy, This group is the focus of majority of previous studies because it is the group at highest risk, and in which a predictor of haemorrhage can potentially change treatment decision. The authors did not clarify why they excluded this group.
Answer: This point is excellent! This study aimed to explore the risk factors of HT in patients without thrombolytic or thrombectomy therapy. We are sorry for the unclear description. Besides, the number of patients in this study who had thrombolytic or thrombectomy treatment was relatively small. In the future, we will investigate the risk factors of HT in patients with thrombolytic or thrombectomy therapy. We have revised the introduction and discussion section as follows:
Introduction section (page 2, line 79): Previous studies have investigated the risk factors of HT in patients with thrombolytic or thrombectomy therapy, but few in patients without thrombolytic or thrombectomy treatment. This study aimed to determine if G/P was connected with HT independently in patients without thrombolytic or thrombectomy therapy.
Discussion section (page 10, line 294): Fourthly, patients with thrombolytic or thrombectomy treatment in this study were excluded. In the future, we will investigate the relationship between G/P and HT in patients with thrombolytic or thrombectomy therapy with a larger sample.
Question 3: The authors did not clarify why they think the glucose/platelet index is better than glucose level or platelet count alone.
Answer: Thank you very much for your careful review and insightful comments. This point is excellent! According to your suggestion, we performed a receiver operating characteristic (ROC) curve to compare the predictive ability of the G/P, glucose, and platelet. The results (Supplement Figure 1) showed that G/P had the best predictive value (AUC: 0.616, 95% CI 0.594-0.637) than glucose (AUC:0.594, 95% CI 0.571-0.616) and platelet (AUC: 0.563, 95% CI 0.541-0.585) (both P < 0.05).
Supplement Figure 1:
Results section (page 9, line 227): Besides, we performed a receiver operating characteristic (ROC) curve to compare the predictive ability of the G/P, glucose, and platelet. The results (Supplement Figure 1) showed that G/P had the best predictive value (AUC: 0.616, 95% CI 0.594-0.637) than glucose (AUC:0.594, 95% CI 0.571-0.616) and platelet (AUC: 0.563, 95% CI 0.541-0.585) (both P < 0.05).
Question 4: Other minor comments
Line 101: smoking and alcohol are lifestyle risk factors for stroke, but not usually referred to as hobbies.
Answer: Thanks very much for such careful reviewing. We have revised “hobbies” into “lifestyle risk factors for stroke”.
Question 5: Line 118: Baseline imaging is reported to be done at a different department. Why is that?
Answer: Thank you very much for your careful review and insightful comments! Acute stroke patients are often treated in the emergency department or another hospital before admission. Then they are sent to our hospital’s inpatient neurology department, so the baseline imaging comes from different departments.
Question 6: And is there any further information on time between onset of symptoms and scans (baseline and follow up).
Answer: Thank you very much for your suggestion. We analyzed time intervala (time from stroke onset to baseline scans) and time intervalb (time from stroke onset to follow-up scans). Results showed that patients with HT had a longer time interval between the onset of stroke and follow-up scans compared with patients without HT [5(3-7), 7(5-11), P<0.001]. We have controlled the time in the logistic regression and the results remained unchanged (OR = 1.85, 95% CI = 1.31-2.63, P < 0.001). In addition, this time interval can still lead to bias, which we have included as a limitation. We have modified the methods, results, and discussion section as follows:
Methods section (page 3, line 112): We collected and documented the demographic data (age and gender), lifestyle risk factors for stroke (drinking and smoking history), baseline clinical parameters [body mass index (BMI), hypertension, diabetes mellitus, AF], laboratory information [glucose levels, platelet count, creatine, prothrombin time (PT), and international normalized ratio (INR)], treatment during hospitalization (antiplatelet, anticoagulation, and statins), the time interval between the onset of stroke and baseline scans, and the time interval between the onset of stroke and follow-up scans.
Results section (page 5, line 170): Results showed that patients with HT had a longer time interval than those without HT [5(3-7) vs. 7(5-11), P < 0.001].
Discussion section (page 10, line 297): Fifthly, we analyzed the time interval between the onset of symptoms and scans (baseline and follow-up). Results showed that patients with HT had a longer time interval between baseline and follow-up imaging than those without HT, which may lead to bias.
Question 7: I note that the study only included patients were a repeat scan was done for clinical reasons. This might be a source of bias as ICH is often asymptomatic. Is it clear whether the heamorrhagic changes were always symptomatic.
Answer: Thank you very much for your valuable advice. We appreciate it very much!
Finally, although we routinely repeated CT or MRI scans, we still could not completely confirm that all the patients with asymptomatic hemorrhagic were identified in this study, which may lead to bias. Thus, more rigorous prospective studies are needed in the future.
Discussion section (page 11, line 300): Finally, although we routinely repeated CT or MRI scans, we still could not completely confirm that all the patients with asymptomatic hemorrhagic were identified in this study, which may lead to bias. Thus, more rigorous prospective studies are needed in the future.

Reviewer 2 Report
In the presented paper the authors highlight glucose/platelet ratio as a new potential biomarker associated with increased risk of hemorrhagic transformation of ischemic stroke.
The paper is well-prepared, written in a scientific sound, with a good English style and grammar nad clear graphics and tables.
The results are very important for a daily clinical practise as a G/P ratio is a cheap and a common test to perform in Stroke Units. Thus, there is great chance for a furher research in this field, to confirm above findings.
I did not find any significant flaws or drawbacks.
Author Response
Responses to the Reviewer 2:
Question 1: In the presented paper the authors highlight glucose/platelet ratio as a new potential biomarker associated with increased risk of hemorrhagic transformation of ischemic stroke. The paper is well-prepared, written in a scientific sound, with a good English style and grammar and clear graphics and tables. The results are very important for a daily clinical practise as a G/P ratio is a cheap and a common test to perform in Stroke Units. Thus, there is great chance for a furher research in this field, to confirm above findings. I did not find any significant flaws or drawbacks.
Answer: Thank you very much for your affirmation!
Reviewer 3 Report
In this work, the authors relate a ratio constructed from the levels of blood glucose and platelets with the risk of suffering a hemorrhagic transformation in an acute cerebral infarction. This is a single-center study but with a more than correct number of patients. The methodology is adequate and well explained, although corrections are necessary in certain expressions that do not fit with the common use of English.
However, I consider that the argumentative basis of the work is not sufficient for its publication. As the authors comment, it is already known that both blood glucose and platelet count influence the risk of hemorrhagic transformation. Nevertheless, the authors do not conjecture a pathophysiological connection between the two elements that justifies creating a ratio between the two. Taking this into account, I don't see how the realization of an artificial ratio relating both values ​​is more interesting than said values ​​separately.
Author Response
Responses to the Reviewer 3:
Question 1: In this work, the authors relate a ratio constructed from the levels of blood glucose and platelets with the risk of suffering a hemorrhagic transformation in an acute cerebral infarction. This is a single-center study but with a more than correct number of patients. The methodology is adequate and well explained, although corrections are necessary in certain expressions that do not fit with the common use of English.
However, I consider that the argumentative basis of the work is not sufficient for its publication. As the authors comment, it is already known that both blood glucose and platelet count influence the risk of hemorrhagic transformation. Nevertheless, the authors do not conjecture a pathophysiological connection between the two elements that justifies creating a ratio between the two. Taking this into account, I don't see how the realization of an artificial ratio relating both values is more interesting than said values separately.
Answer: Thank you very much for your careful review and insightful comments. This point is excellent! According to your suggestion, we performed a receiver operating characteristic (ROC) curve to compare the predictive ability of the G/P, glucose, and platelet. The results (Supplement Figure 1) showed that G/P had the best predictive value (AUC: 0.616, 95% CI 0.594-0.637) than glucose (AUC:0.594, 95% CI 0.571-0.616) and platelet (AUC: 0.563, 95% CI 0.541-0.585) (both P < 0.05).
Supplement Figure 1:
Results section (page 9, line 227): Besides, we performed a receiver operating characteristic (ROC) curve to compare the predictive ability of the G/P, glucose, and platelet. The results (Supplement Figure 1) showed that G/P had the best predictive value (AUC: 0.616, 95% CI 0.594-0.637) than glucose (AUC:0.594, 95% CI 0.571-0.616) and platelet (AUC: 0.563, 95% CI 0.541-0.585) (both P < 0.05).

Reviewer 4 Report
This was a retrospective study aimed to explore the relationship between glucose to platelet ratio andhemorrhagic transformation (HT) in acute ischemic stroke.
The study is overall interesting and provide useful information for clinical practice. There are, however, some issues that need to be further addressed.
One possible mechanism linking high glucose levels with HT is inflammation. In this regard, it should be acknowledged how inflammatory biomarkers have been already associated with HT and stroke outcome (Ref. Neutrophil-to-Lymphocyte Ratio and Symptomatic Hemorrhagic Transformation in Ischemic Stroke Patients Undergoing Revascularization. Brain Sci 2020; Systemic Inflammatory Response Index and Futile Recanalization in Patients with Ischemic Stroke Undergoing Endovascular Treatment. Brain Sci 2021; Early ficolin-1 is a sensitive prognostic marker for functional outcome in ischemic stroke. J Neuroinflammation. 2016).
Further, there is increasing evidence that easily available serum biomarkers of inflammation can be reliable predictors of outcome also in patients with ICH and improve the outcome prediction when added to validated prognostic scales (Ref. Monomeric C-Reactive Protein and Cerebral Hemorrhage: From Bench to Bedside. Front Immunol 2018; Matrix Metalloproteinases in Acute Intracerebral Hemorrhage. Neurotherapeutics 2020; Neutrophil-to-lymphocyte ratio improves outcome prediction of acute intracerebral hemorrhage. J Neurol Sci 2018). In view of this evidence, it would be fine to discuss briefly how common inflammatory pathways may underlying secondary brain damage in both ischemic and hemorrhagic stroke.
The predictive accuracy of SIRI should be compared with the one of its components alone, i.e., neutrophils, lymphocytes, monocytes to provide evidence about the net clinical benefit to use this composite marker.
Author Response
Responses to the Reviewer 4:
Question 1: This was a retrospective study aimed to explore the relationship between glucose to platelet ratio andhemorrhagic transformation (HT) in acute ischemic stroke.
The study is overall interesting and provide useful information for clinical practice. There are, however, some issues that need to be further addressed. One possible mechanism linking high glucose levels with HT is inflammation. In this regard, it should be acknowledged how inflammatory biomarkers have been already associated with HT and stroke outcome (Ref. Neutrophil-to-Lymphocyte Ratio and Symptomatic Hemorrhagic Transformation in Ischemic Stroke Patients Undergoing Revascularization. Brain Sci 2020; Systemic Inflammatory Response Index and Futile Recanalization in Patients with Ischemic Stroke Undergoing Endovascular Treatment. Brain Sci 2021; Early ficolin-1 is a sensitive prognostic marker for functional outcome in ischemic stroke. J Neuroinflammation. 2016). Further, there is increasing evidence that easily available serum biomarkers of inflammation can be reliable predictors of outcome also in patients with ICH and improve the outcome prediction when added to validated prognostic scales (Ref. Monomeric C-Reactive Protein and Cerebral Hemorrhage: From Bench to Bedside. Front Immunol 2018; Matrix Metalloproteinases in Acute Intracerebral Hemorrhage. Neurotherapeutics 2020; Neutrophil-to-lymphocyte ratio improves outcome prediction of acute intracerebral hemorrhage. J Neurol Sci 2018). In view of this evidence, it would be fine to discuss briefly how common inflammatory pathways may underlying secondary brain damage in both ischemic and hemorrhagic stroke.
Answer: Thank you for your valuable comment! We have added the inflammation to the discussion section as follows:
Discussion section (page 9, line 255): Thirdly, inflammation is exacerbated by the production of proinflammatory cytokines, apoptosis, and the exacerbation of cytotoxic edema when glucose levels are high [28, 31]. Inflammation plays a vital role in HT. The neutrophil-to-lymphocyte ratio can accurately predict symptomatic hemorrhagic transformation in AIS patients who undergo revascularization [32]. AIS patients having endovascular treatment and achieving effective recanalization with a higher systemic inflammatory response index at admission had an increased risk of a poor functional outcome at three months [33]. Early ficolin-1 is one of the most sensitive predictors of functional prognosis in AIS [34]. Further, there is increasing evidence that readily available serum biomarkers of inflammation can also be reliable predictors of outcomes in patients with ICH and improve the outcome prediction when added to validated prognostic scales. C-reactive protein, an important inflammatory marker, is associated with intracerebral hemorrhage outcomes [35]. Matrix metalloproteinases have a pleiotropic and biphasic effect on acute intracerebral hemorrhage [36]. Besides, the neutrophil-to-lymphocyte ratio is associated with 30-day functional status after acute intracerebral hemorrhage [37].
Question 2: The predictive accuracy of G/P should be compared with one of its components alone, to provide evidence about the net clinical benefit to use this composite marker.
Answer: Thank you very much for your careful review and insightful comments. This point is excellent! According to your suggestion, we performed a receiver operating characteristic (ROC) curve to compare the predictive ability of the G/P, glucose, and platelet. The results (Supplement Figure 1) showed that G/P had the best predictive value (AUC: 0.616, 95% CI 0.594-0.637) than glucose (AUC:0.594, 95% CI 0.571-0.616) and platelet (AUC: 0.563, 95% CI 0.541-0.585) (both P < 0.05).
Supplement Figure 1:
Results section (page 9, line 227): Besides, we performed a receiver operating characteristic (ROC) curve to compare the predictive ability of the G/P, glucose, and platelet. The results (Supplement Figure 1) showed that G/P had the best predictive value (AUC: 0.616, 95% CI 0.594-0.637) than glucose (AUC:0.594, 95% CI 0.571-0.616) and platelet (AUC: 0.563, 95% CI 0.541-0.585) (both P < 0.05).
Reference
28. Yang, C., et al., Neuroinflammatory mechanisms of blood-brain barrier damage in ischemic stroke. Am J Physiol Cell Physiol, 2019. 316(2): p. C135-C153.
31. Salman, M., et al., Acute Hyperglycemia Exacerbates Hemorrhagic Transformation after Embolic Stroke and Reperfusion with tPA: A Possible Role of TXNIP-NLRP3 Inflammasome. J Stroke Cerebrovasc Dis, 2022. 31(2): p. 106226.
32. Switonska, M., et al., Neutrophil-to-Lymphocyte Ratio and Symptomatic Hemorrhagic Transformation in Ischemic Stroke Patients Undergoing Revascularization. Brain Sci, 2020. 10(11).
33. Lattanzi, S., et al., Systemic Inflammatory Response Index and Futile Recanalization in Patients with Ischemic Stroke Undergoing Endovascular Treatment. Brain Sci, 2021. 11(9).
34. Zangari, R., et al., Early ficolin-1 is a sensitive prognostic marker for functional outcome in ischemic stroke. J Neuroinflammation, 2016. 13: p. 16.
35. Di Napoli, M., et al., Monomeric C-Reactive Protein and Cerebral Hemorrhage: From Bench to Bedside. Front Immunol, 2018. 9: p. 1921.
36. Lattanzi, S., et al., Matrix Metalloproteinases in Acute Intracerebral Hemorrhage. Neurotherapeutics, 2020. 17(2): p. 484-496.
37. Lattanzi, S., et al., Neutrophil-to-lymphocyte ratio improves outcome prediction of acute intracerebral hemorrhage. J Neurol Sci, 2018. 387: p. 98-102.

Round 2
Reviewer 1 Report
The authors addressed all raised concerns in initial review.
Reviewer 3 Report
Although I thank the authors for their response and the effort they have put into responding to my review, I continue to consider that the realization of this ratio (although it has statistical relevance) does not have the necessary pathophysiological bases to be considered relevant in usual clinical practice.
Reviewer 4 Report
The Authors addressed all the issues